# Water Uptake Pattern by Coniferous Forests in Two Habitats Linked to Precipitation Changes in Subtropical Monsoon Climate Region, China

**Jianbo Jia** [1,2,3] **, Yu Chen** [1,2] **, Jia Lu** [1,2] **and Wende Yan** [1,3,*]

1   Central South University of Forestry and Technology, Changsha 410004, China; jotham880303@163.com (J.J.); cchenyu0702@163.com (Y.C.); t20162294@csuft.edu.cn (J.L.)
2   Key Laboratory of Soil and Water Conservation and Desertification Combating in Hunan Province, Changsha 410004, China
3   Lutou National Station for Scientific Observation and Research of Forest Ecosystem in Hunan Province, Yueyang 414000, China
*   Correspondence: csfuywd@hotmail.com

**Abstract:** Variations in precipitation patterns under climate changes influence water availability, which has important implications for plants' water use and the sustainability of vegetation. However, the water uptake patterns of the main forest species under different temporal spatial conditions of water availability remain poorly understood, especially in areas of high temporal spatial heterogeneity, such as the subtropical monsoon climate region of China. We investigated the water uptake patterns and physiological factors of the most widespread and coniferous forest species, *Cunninghamia lanceolata* L. and *Pinus massoniana* L., in the early wet season with short drought (NP), high antecedent precipitation (HP), and low antecedent precipitation (LP), as well as in the early dry season (DP), in edaphic and rocky habitats. The results showed that the two species mainly absorbed soil water from shallow layers, even in the short drought period in the wet season and switched to deeper layers in the early dry season in both habitats. It was noted that the trees utilized deep layers water in edaphic habitats when the antecedent rainfall was high. The two species showed no significant differences in water uptake depth, but exhibited notably distinct leaf water potential behavior. *C. lanceolata* maintained less negative predawn and midday water potential, whereas *P. massoniana* showed higher diurnal water potential ranges. Moreover, the water potential of *P. massoniana* was negatively associated with the antecedent precipitation amount. These results indicate that for co-existing species in these communities, there is significant eco-physiological niche segregation but no eco-hydrological segregation. For tree species in two habitats, the water uptake depth was influenced by the available soil water but the physiological factors were unchanged, and were determined by the species' genes. Furthermore, during the long drought in the growing season, we observed probable divergent responses of *C. lanceolata* and *P. massoniana*, such as growth restriction for the former and hydraulic failure for the latter. However, when the precipitation was heavy and long, these natural species were able to increase the ecohydrological linkages between the ecosystem and the deep-layer system in this edaphic habitat.

**Keywords:** plant water source; habitat; stable isotope technology; leaf water potential; water use efficiency

## 1. Introduction

Increases in vegetation greenness have been reported around the world over the last three decades, manifested as the expansion of afforestation and reforestation [1–3]. However, forests may be vulnerable to degradation due to global climate changes with new precipitation patterns [4–6]. Changes in the characteristics of precipitation may result in changes in water availability, which have implications for plants' water uses in ecosystems [7,8]. The variations in plants' water use responses to precipitation and water

availability plays important role in the sustainability of the restored vegetation and the promotion of the water cycle in critical zones [9–11].

The temporal–spatial heterogeneity of precipitation and water availability affect plant water use strategies [12,13]. At the point scale, the water source variability along the soil profile is one of the most important factors for water uptake by plants [14]. At the surface scale, the aquifer storage is distinct in different habitats, such as deep soil habitats [15], outcrop habitats [16], and soil with rock fragments habitats [17], which is related to the soil properties and plant water consumption. At different stages of the same season, the plant water uptake depth may also differ with changes in rhizosphere water availability [15,18]. Meanwhile, the amount of precipitation may be a critical factor affecting the water sources of trees. The plant water uptake can be identified by contrasting the $\delta D$ and $\delta^{18}O$ of xylem water and all the potential water sources. Previous studies have shown that tree species may switch their water sources from shallow layers in the wet season with sufficient precipitation to stable layers in the dry season using stable isotope techniques [19–21] Liu [22] found that following rainfall events, *Platycladus orientalis* L. trees with a dense and shallow fine root system absorbed more water from the soil surface layers and precipitation. Other plants, however, mostly take up water from deep and stable layers regardless of seasonal changes or precipitation events in the semi-arid regions [16,23]. In contrast, in subtropical regions, evergreen species use shallow soil water with a drought-avoidance strategy even under seasonal drought conditions [24].

The divergent response of plant water uptake to changes in precipitation and water availability has been related to physiological characteristics. It has been suggested that the predawn and midday leaf water potential can be used to describe the daily patterns of plant-water relations, coupling water among the root zone, the plant itself and the atmosphere [25]. Previous studies have shown that plants relying on shallower water sources exhibited a larger diurnal range of leaf water potential, and on the contrary, narrower diurnal ranges are usually linked with deep and stable water sources [26–28]. Moreover, the plant water efficiency (WUE) has attracted attention as a means of reflecting plant water use characteristics, together with plant water uptake [29,30]. Nie [31] explored leaf WUE based on $\delta^{13}C$ values and found that the high WUE corresponded with the use of deep water sources, indicating more conservative water-use strategies in a subtropical monsoon climate region. The plant water uptake pattern was found to be influenced by water availability and physiological traits in different ecosystems [32]. However, the relationship between these two factors affecting plant water uptake is unclear, especially in complex and fragile forest ecosystems, which limits the understanding of restored vegetation adaptability and rock-soil-water-plant-atmosphere interactions in critical zones.

Subtropical China, which is characterized by a monsoon climate, is an ecologically sensitive area that is affected by global changes [33]. The precipitation in this region is abundant and the alternation of dry and wet is obvious. The change in precipitation patterns has led to a reduction in the available water in the ecosystem, and the risk of drought stress and drought death has significantly increased [4]. The distribution of the plantations is a clustered distribution with heterogeneous habitats (such as thin soil habitats with rock fragments, and outcrop habitats with soil fragments) [34]. Different rock and soil structures could influence hydrological processes and the amount of soil water available. Plant water use strategies in different habitats are critically important for the evaluation of vegetation adaptation. A number of previous studies have primarily focused on the water sources of different types of plantations or natural vegetation in one specific habitat [16,35]. Few have paid attention to differences in plant water uptake patterns in different habitats, which has limited our understanding of plant water adaptation and the evaluation of sustainable vegetation restoration. With changes in the global precipitation pattern, short-term drought and rainstorms have become more frequent, especially in the wet/growing season. However, it is unclear how the water uptake of plants in the different habitats responds to these precipitation changes.

Based on the above analysis, we applied stable isotope techniques ($\delta$D and $\delta^{18}$O) and measurements of leaf water potential to determine coniferous-leaved forest water uptake patterns in two habitats (edaphic and rocky habitats) with different antecedent precipitation during the growing season in the subtropical monsoon climate region of China. The main objectives of this study were: (i) to investigate the responses of belowground water use patterns of *Cunninghamia lanceolata* L. and *Pinus massoniana* L. to the temporal–spatial heterogeneity of water availability, as shown in conditions of different antecedent precipitation levels and edaphic and rocky habitats; and (ii) to understand the aboveground physiological responses to varied water availability of two species, analyzed by examining the variations in leaf water potential behavior and water use efficiency. Our first hypothesis was that soil water availability could have an effect on the plant water uptake depth, and that the two species may show similar water sources, and the second was that the plants' physiological factors may vary with the changes in water availability and species types.

## 2. Materials and Methods

### 2.1. Study Sites

This study was conducted at the Hunan Lutou forest ecosystem observation and research Station (28°31′7″–28°38′ N, 113°51′52″–113°58′24″ E) (Figure 1). The region has a subtropical mountainous monsoon climate, with a mean annual precipitation of 1450.8 mm and an annual temperature of 18.5 °C. The wet season, which receives more than 60% of the annual rainfall, lasts from late April to late September, and the dry season extends from December to February [28]. The growing season spans from April to October. The study area is dominated by *Cunninghamia lanceolata* and *Pinus massoniana* secondary forests. The understory contains species such as *Fortunearia sinensis* Rehd, *Ilex cornuta* and *Asparagus cochinchinensis*, and the forest coverage rate is more than 90%. Soil in the study area is predominantly red soil, having a general soil layer that is 80–100 cm thick. The other part of the slope has a high exposed rock ratio whereas the soil occurs discontinuously, only in rock gaps. Thus, the habitats were variable, with the different outcrop ratios, such as an edaphic habitat with a low outcrop ratio, a continuous broken rock habitat with patches of soil, an isolated outcrop habitat, and so on. Springs sometimes appear at the bottom of hillslopes during the rainy season or after rains in the drought season.

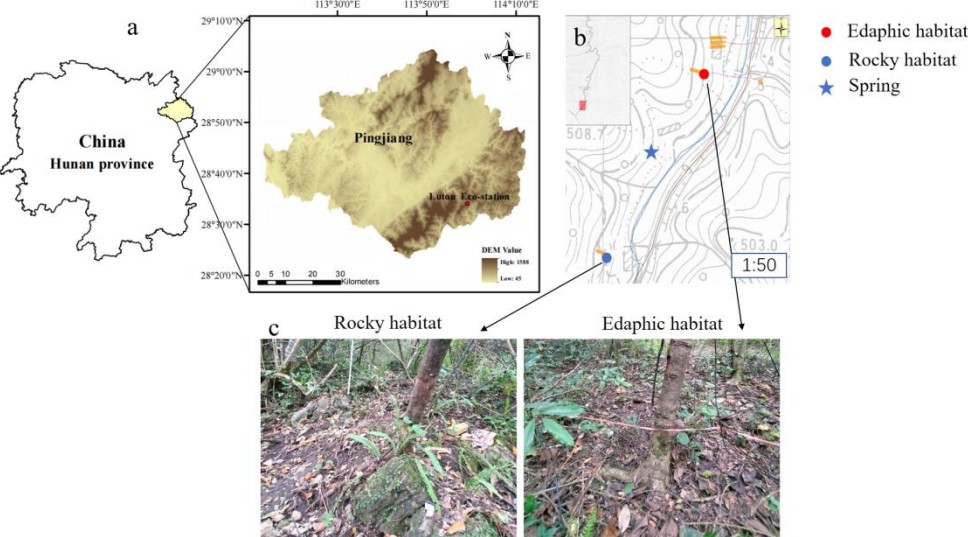

**Figure 1.** Map of the study and the field area. (**a**): The location of Lutou forest ecosystem observation and research Station in Hunan province; (**b**): The sampling site location in the study area; (**c**) photographs of the two habitats.

According to the distribution of these typical habitats, one habitat consisting of thick soil with rock fragments habitat (the "edaphic habitat" for short) and the another habitat

consisting of continuous stone outcrops with soil fragments (the "rocky habitat" for short) were chosen at the foot of the Northwest-facing hillslope in two 20 m × 20 m sample plots. The two habitats were 250 m apart whereas the elevation difference was about five meters. In the edaphic habitat, the soil was relatively thick (about 90 cm deep), horizontally interrupted by small outcrops, and vertically interrupted by small rocks. Along the soil profile, the upper layer soil (0–30 cm) was well-drained whereas the lower layers (30–70 cm) were sticky with a low saturated hydraulic conductivity ($K_s$). Underneath the soil was a high-weathered dolomite bedrock zone (70–90 cm). The outcrop ratio was about 20% in this habitat. In the rocky habitat, the outcrop ratio was more than 80%, and the range of height from the top of the outcrop to the soil in the rock gaps was from 0.3 m to 3 m. The soil was inlaid in the rock in a spotty pattern and was discontinuous (average 30 cm deep on average). Similarly, a high-weathered bedrock zone was present under the soil. The vegetation was sparse in this habitat. There was an intermittent spring outflow near the two habitats at the bottom of the hillslope.

For the subsequent analysis and comparison, the plant water sources were divided into shallow (0–30 cm), middle (30–70 cm in the edaphic habitat and 30–50 cm in the rocky habitat), and deep (70–90 cm in the edaphic habitat and 50–70 cm in the rocky habitat) layers and spring according to the soil texture and fluctuations and patterns of isotopic ratios in the soil water, VWC, and the impact of the rainfall pulse. (1) Shallow soil layer: The variability of soil water isotopic compositions and VWC in this layer was larger, and it was vulnerable to rainfall pulses and evaporation with seasons. (2) Middle soil layer: The variability of soil water isotopic compositions and SWC in this layer was lower than that of the 0–30 cm soil layer. The impacts of rainfall pulses and evaporation were moderate. Both the clay content and soil bulk density were higher than the shallow layers. (3) Deep soil layer: This layer was high-weathered bedrock with high leakage and low water holding capacity in the rocky habitat and high water moisture in the edaphic habitat, respectively.

*2.2. Plant and Soil Sampling*

In order to explore the relationship between plant physiological traits and water uptake patterns for adapting to precipitation change, plant and soil sampling were conducted simultaneously at the two habitats bimonthly on 12 June (wet season with high antecedent precipitation, HP), 5 August (wet season with low antecedent precipitation, LP), and 18 October (early dry season, DP) 2020. We also sampled on 18 May in the early wet season with a 20 day drought (no rain, NP). Two coniferous species, *Cunninghamia lanceolata* (DBH of from 5 to 11cm, average DBH was 7.9 cm) and *Pinus massoniana* (DBH of from 6 to 12cm, average DBH was 8.5cm) in each of the habitats were selected for the study. We selected four individuals per species for analysis, and the DBH of sampled trees were used to represent the average DBH in the stands. The leaf and plant xylem samples from every selected plant were collected in each habitat. Every selected plant was collected in each stand-age tree per month. The fully sun-exposed, mature and healthy leaves in the upper canopy from each selected plant were collected in different directions on each sampling date. The leaves were mixed and packed into craft paper bags and brought them back to the laboratory for the measurement of the plant leaves' $\delta^{13}C$ levels. Shoots ranging from 0.3 to 0.5 cm in diameter and 3 to 5cm in length were collected at mid-day from stems that were more than 2 years old [28]; the outer bark and phloem of the shoots were removed to obtain the xylem sample.

Soil samples were obtained in two habitats from six depth intervals (0–10, 10–20, 20–30, 30–50, 50–70, 70–90 cm) with an auger (sampling only at 70 cm deep in the rocky habitat) and five replicates were collected at each layer. Among them, the high-weathered bedrock samples were collected between 70–90 cm in the edaphic habitat and 50–70 cm in the rocky habitat. A subsample of the soil samples was stored at −20 °C for isotopic analysis, whereas the remainder of the samples were sealed for the measurement of gravimetric soil water content, obtained by oven drying for one day. The volumetric water content (VWC) was converted according to gravimetric water content and bulk density (Table 1) of each layer.

**Table 1.** The soil bulk density of two the habitats.

| Soil Depth (cm) | Edaphic Habitat (g·cm$^{-3}$) | Rocky Habitat (g·cm$^{-3}$) |
|:---:|:---:|:---:|
| 0–10 | 0.90 | 0.88 |
| 10–20 | 0.97 | 0.85 |
| 20–30 | 1.09 | 0.91 |
| 30–50 | 1.13 | 1.03 |
| 50–70 | 1.01 | 0.88 |
| 70–90 | 0.91 | - |

*2.3. Precipitation and Spring Sampling*

Water samples were routinely collected for each rain event above 5 mm from May 2020 to December 2020. The isotopic values of precipitation were not collected from January to April due to the COVID-19 pandemic. The collection equipment was designed based on the new device for monthly rainfall sampling developed for the Global Network of Isotopes in Precipitation [36]. The rainwater samples were stored in cap vials, wrapped in parafilm, and stored in a freezer until the analysis of stable isotopes. Data on temporal distribution of rainfall data and other meteorological data were collected at a meteorological station located in the middle of the same small catchment. Spring water discharged from 1 June to 29 November, but were cut off between 25 July to 29 August. The spring was sampled regularly during the outflow period. Both rainwater and spring water were stored in cap vials, wrapped in parafilm, and frozen until stable isotope analysis.

*2.4. Isotopic Analyses*

The water was extracted from the xylem and soil using an automatic cryogenic vacuum distillation water extraction system (LI-2100, LICA, Beijing, China) [37,38]. The $\delta D$ and $\delta^{18}O$ in the xylem and soil water samples were measured with liquid water isotope ratio infrared spectroscopy (IRIS, DLT-100, Los Gatos Research, Mountain View, CA, USA) at the Key Laboratory for Agro-Ecological Processes in Subtropical Region, Chinese Academy of Sciences. The $\delta^{13}C$ level in the plant leaves were analyzed using an isotope ratio mass spectrometer (IRMS, MAT253, Thermo Fisher Scientific, Bremen, Germany).

The isotope composition is reported in $\delta$ notation relative to V-SMOW as

$$\delta X = (R_{sample}/R_{standard} - 1) \times 1000 \tag{1}$$

where X represents D, $^{18}O$, or $^{13}C$. $R_{sample}$ and $R_{standard}$ are the ratios of D/H, $^{18}O/^{16}O$, or $^{13}C/^{12}C$ ratio of a measured sample and a standard sample, respectively. The standard deviation for repeat measurements was $\pm 1‰$ for $\delta D$, $\pm 0.2‰$ for $\delta^{18}O$, and $\pm 0.15‰$ for $\delta^{13}C$.

Extracting water from the plant xylem using cryogenic vacuum distillation can result in the mixing of organic materials (e.g., methanol and ethanol), which may affect the spectroscopy and lead to erroneous stable isotope values when analyzing them with IRIS [39,40]. We have corrected the isotopic values of the xylem according to Liu [28].

*2.5. Leaf Water Potential*

Predawn and midday water potentials ($\Psi_{pd}$ and $\Psi_{md}$, respectively) of leaves were measured in the wet seasons (simultaneously with isotope sampling) with a pressure chamber (PMS Instruments Co., Corvallis, OR, USA). Samples (*n* = 5 per species) were collected from branches that were fully exposed to the sun, at places where branches were 2/3 of the way up of the canopy, at least 2 m above ground. The measurements were performed between 4:00 to 6:00 h for predawn water potential and between 12:00 and 14:00 h for midday water potential on the same day.

*2.6. Data Analysis*

Plant water source partitioning was determined by means of the Bayesian mixing model MixSIAR (version 3.1.7) [41]. The raw isotopic ratios of the xylem water were input into MixSIAR as the mixture data. The averages and standard deviations of the soil water isotopes in the different soil layers were the source data. The discrimination was set to zero for both δD and δ¹⁸O because there is generally no isotopic discrimination of water during plant water uptake by roots [42].

Independent-samples *t*-tests and One-way ANOVA were used to detect the differences in plant water sources and water potential among the species, habitats and their seasonal differences. Post hoc comparisons were based on Tukey's HSD. Moreover, Pearson correlation was used to conduct the correlation analysis, and the figures were plotted with Origin 9.0 software (Origin, Origin Lab, Farmington, ME, USA).

## 3. Results

*3.1. Isotopic Compositions of Precipitation, Soil Water, and Springs*

The total precipitation was approximately 2121 mm in 2020 and the distribution of rainfall was temporally uneven, with 79.32% of the rainfall occurring during the wet season (Figure 2). It was noted that there are two extreme precipitation events occurred—on 7 in September (282.2 mm) and on 7 June (115.2 mm). Except for the NP sampling with a 20 day drought, the accumulated precipitation amounts ten days before the last three samplings were 283.6 mm, 49.4 mm, and 55.4 mm, respectively.

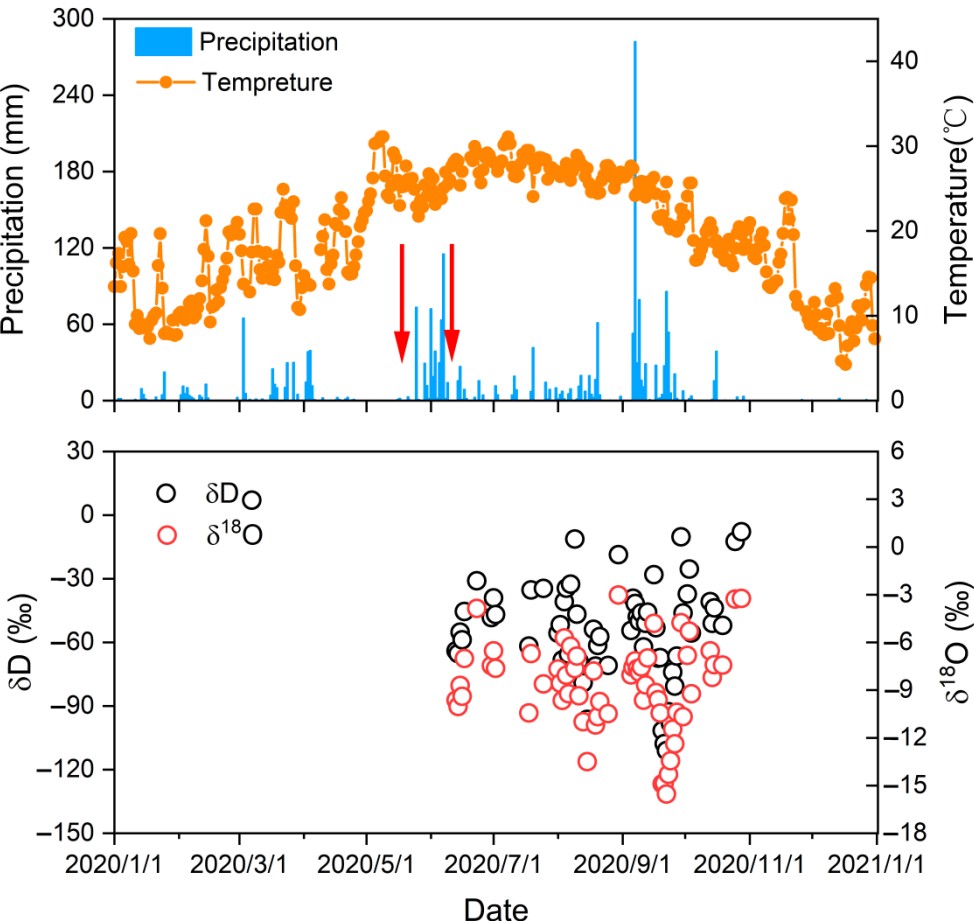

**Figure 2.** Variations in precipitation, mean air temperature, and isotopic values (δD, δ¹⁸O) in precipitation at a daily timescale in 2020. Arrows indicate sampling dates. (The isotopic values of precipitation were not collected from January to April due to the COVID-19 pandemic impacting).

The isotopic compositions of the precipitation showed a large fluctuation during the study period (Figure 2). The mean $\delta$D of the precipitation was $-48.69$ ‰, the mean $\delta^{18}$O of the precipitation was $-7.88$ ‰. The relatively depleted isotopic values of precipitation occurred when it rained continuously for a long time, with high precipitation. The $\delta$D level obtained for ten days of precipitation before the three samplings were ranged from $-23.55$‰ to $-57.52$‰, $-34.54$‰ to $-68.36$‰, $-40.76$‰ to $-51.02$‰, respectively. The $\delta^{18}$O of precipitation before three samplings were ranging from $-5.27$‰ to $-8.15$ ‰, $-7.6$ ‰ to $-9.65$ ‰, $-6.54$‰ to $-7.4$‰, respectively.

The $\delta$D and $\delta^{18}$O values of soil water in the different habitats varied with soil depth and season (Figures 3 and 4). In the edaphic habitat, the average $\delta$D value of the soil water was $-45.56$‰ $\pm$ 16.05 ‰ (mean $\pm$ S.D.), and the average $\delta^{18}$O value was $-6.55$‰ $\pm$ 1.73‰. The average $\delta$D and $\delta^{18}$O values of soil water in the rocky habitat were $-44.6$‰ $\pm$ 16.58‰ and $-6.7$‰ $\pm$ 1.96 ‰, respectively. There were no significant differences ($p = 0.84$ for $\delta$D, $p = 0.79$ for $\delta^{18}$O) in the soils' isotopic compositions in the different habitats. In NP, the soil water isotopes were observed to be depleted with soil depth (Figures 3a and 4a). In HP, the $\delta$D and $\delta^{18}$O values of water along the soil profile were consistent with recent rainfall values (Figures 3b and 4b). In the two late two samplings, the soil water isotope composition converged at the top and bottom layers, which were similar to recent rainfall values (Figure 3c,d, and Figure 4c,d). The middle-layer soil water showed more enriched values in LP and depleted isotopic values in DP and exhibited less variation with soil depth. There were no significant differences obtained for soil water ($p = 1.28$ for $\delta$D, $p = 0.93$ for $\delta^{18}$O) in different sample layers (i.e., 0–10 cm, 10–30 cm, 30–50 cm, 50–70 cm, 70-90 cm). However, when merging sample layers into shallow, middle and deep layers (see Methods), the soil water isotope was significant different in the two habitats ($p < 0.05$).

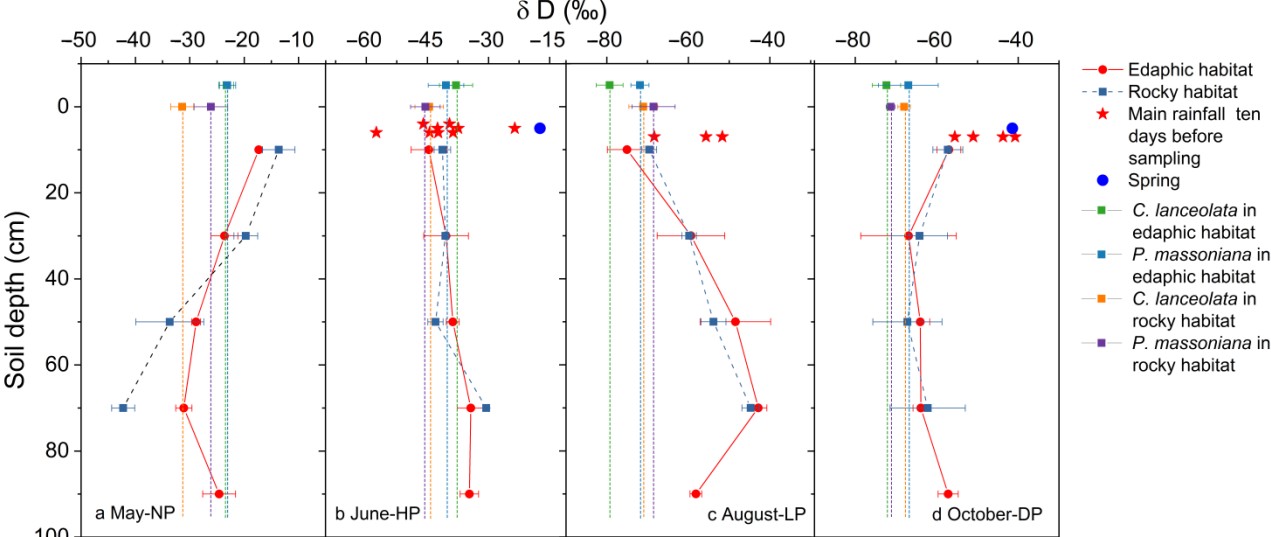

**Figure 3.** Variation in mean ($\pm$S.D.) $\delta$D of soil water with the soil profile, precipitation, spring, and xylem water during the wet season (**a**) May sampling; (**b**) June sampling; (**c**) August sampling; (**d**) October sampling.

The isotopic composition of springs changed across the sampling time. The isotopic values were less negative in HP than in DP. The $\delta$D and $\delta^{18}$O values of xylem water were less negative in NP and became more negative with the seasonal changes. There were no significant differences in isotopic composition between species types and habitats ($p > 0.05$), except in June-HP, when the xylem water isotope was more negative in the rocky habitat than that in the edephic habitat ($p < 0.05$).

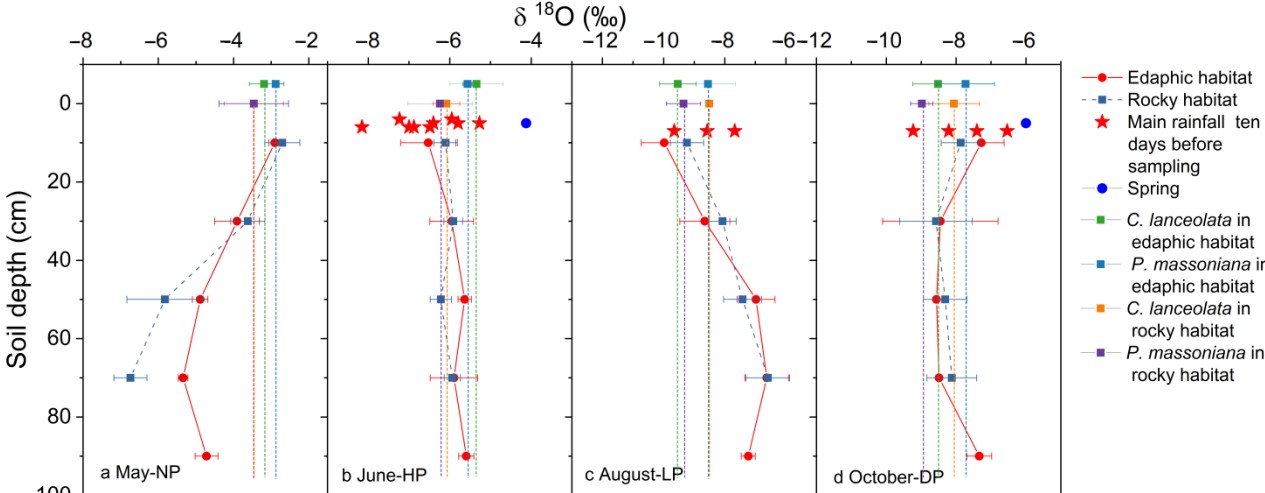

**Figure 4.** Variation in mean (±S.D.) $\delta^{18}$O of soil water with the soil profile, precipitation, spring, and xylem water during the wet season (**a**) May sampling; (**b**) June sampling; (**c**) August sampling; (**d**) October sampling.

### 3.2. Variations in Soil Water Content, Water Uptake Patterns, and Their Linkage with Precipitation

The VWCs of the two habitats displayed clear vertical and seasonal variations (Figure 5). The average VWCs were 43.42% ± 7.68% in the edaphic habitat and 38.24% ± 8.42% in the rocky habitat, with no significant differences ($p = 0.07$) during the study periods. However, the VWCs of shallow soil layers in the two habitats differed significantly ($p < 0.001$). In NP, the VWC of the shallow layer was the lowest in the two habitats and the soil moisture increased with depth (Figure 5a). Furthermore, the VWC exhibited a slightly increasing tendency along the soil profile in the edaphic habitat but a decreasing tendency in the rocky habitat with the seasonal changes. It was noted that the soil moisture in the edaphic habitat was significantly higher than that in the rocky habitat in LP, especially in the middle layers, which may be related to the different soil texture and plant transpiration characteristic (Figure 5c).

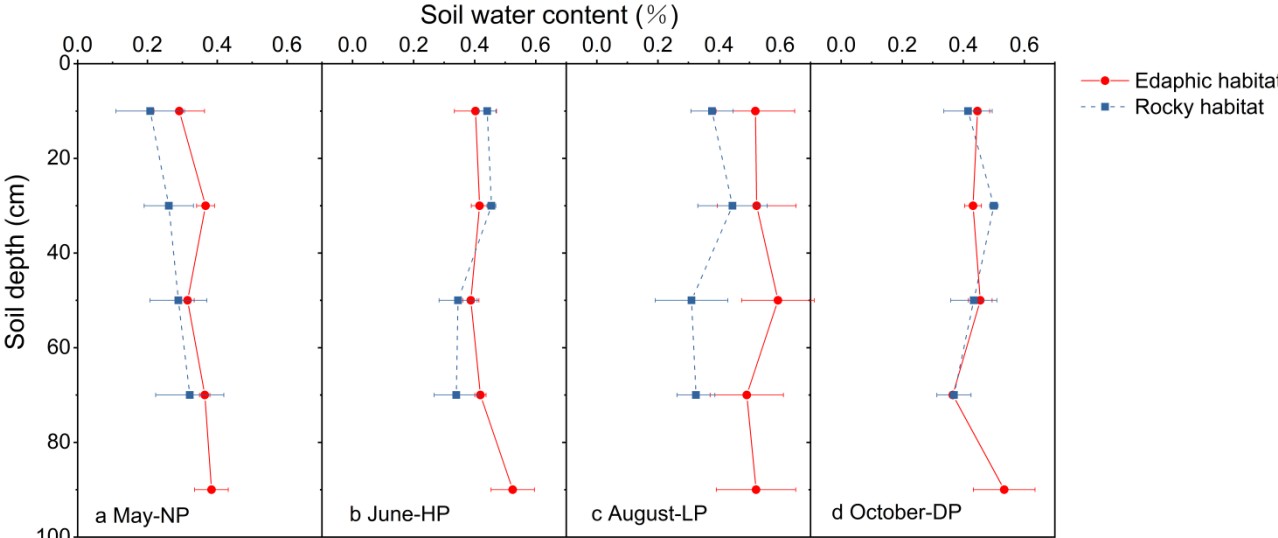

**Figure 5.** Variation in mean (±S.D.) soil water content along the soil profile during the wet season (**a**) May sampling; (**b**) June sampling; (**c**) August sampling; (**d**) October sampling.

The two tree species mainly took up soil moisture throughout the wet season in two habitats, and the proportions of water sources used by the two species exhibited no significant differences ($p > 0.05$) (Figure 6). However, the plant water uptake depth differed between habitats and across seasons. In HP, trees in the rocky habitat absorbed more than 67.14% of their water from shallow soil layers, whereas the mean water uptake ratio of the two tree species in the edaphic habitat were 64.45% for the middle and deep soil layers. In DP, the *C. lanceolata* and *P. massoniana* in edaphic habitat obtained more than 74.82% of their water from the shallow and deep soil layers. On the other hand, in the rocky habitat, the two species mainly extracted soil water from shallow and middle layers (82.13%). In NP and LP, both *C. lanceolata* and *P. massoniana* in the two habitats utilized the largest proportion of shallow soil water (64.97%, 0–30cm).

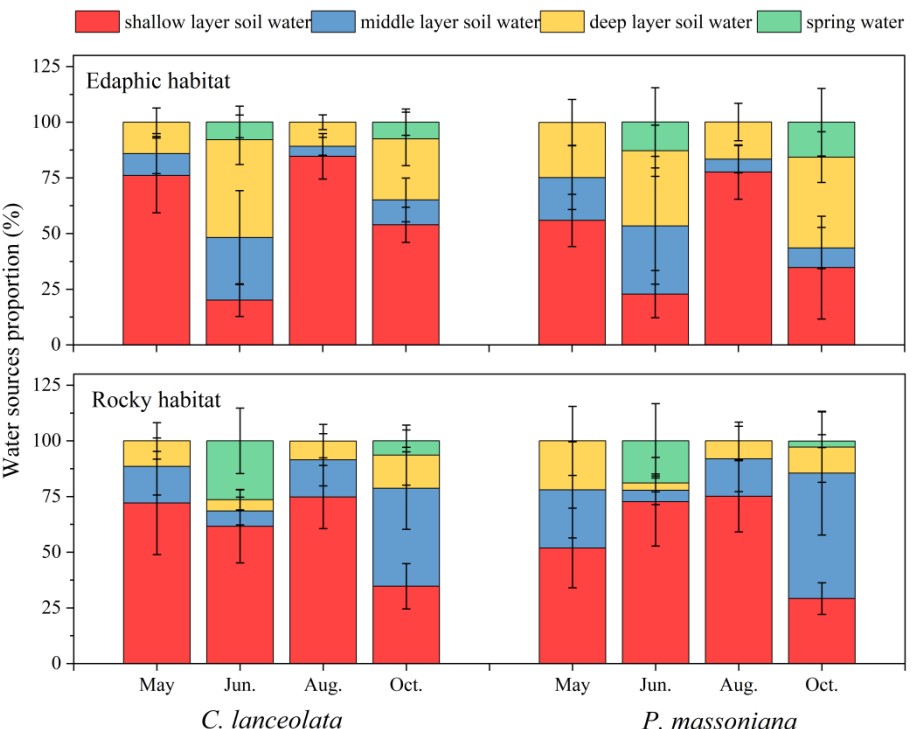

**Figure 6.** Variation in mean (±S.D.) water source proportions for *C. lanceolata and P. massoniana* during the wet season.

The responses of the proportion of plant water sources used in each soil layers to the amount of precipitation ten days before sampling were distinct in the two habitats (Figure 7). In the edaphic habitat, tree species absorbed less water from shallow layers and absorbed more deep soil water with the increases in precipitation (Figure 7a,c). On the other hand, the trees maintained a high water uptake from shallow layers in the rocky habitat regardless of precipitation variations (Figure 7d). Meanwhile, there were significant negative linear relationships between the water source proportions of the middle and deep soil layers and precipitation (Figure 7e,f).

### 3.3. Variation in Leaf Water Potential and Its Linkage with Precipitation

The $\psi_{pd}$ of the two species was found to be less negative ($>-1$MPa) in the two habitats, indicating no severe water stress during the study period, whereas the $\psi_{md}$ was lower than $\psi_{pd}$, away from the 1:1 line, and exhibited profoundly seasonal variations ($p < 0.01$) (Figure 8). Both of the two species showed lower $\psi_{md}$ values in NP and DP than that in HP and LP ($p < 0.05$). Furthermore, *P. massoniana* showed significantly more negative $\psi_{md}$ values than *C. lanceolata*, especially in NP and DP ($p < 0.05$). However, there were no significant differences in $\psi_{pd}$ and $\psi_{md}$ between the edaphic and rocky habitats for the two species ($p > 0.05$).

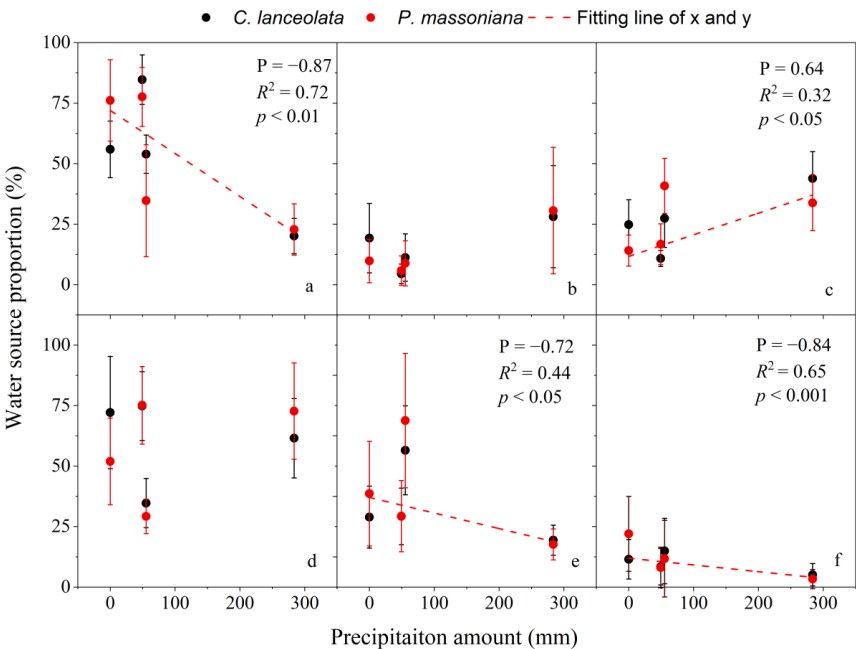

**Figure 7.** Relationships between water source proportions for each soil layers (mean ± S.D.) and the precipitation amount ten days before sampling. P is Pearson correlation, $R^2$ represents the fitting degree of the relationship between the water source proportion and the precipitation amount; $p$ is the $p$-value of the fitting ((**a–c**) plant water sources from the shallow, middle, and deep layers in the edaphic habitat, respectively; (**d–f**) plant water sources from the shallow, middle, and deep layers in the rocky habitat, respectively). The black dots represented *C. lanceolata*, and the red dots represent *P. massoniana*.

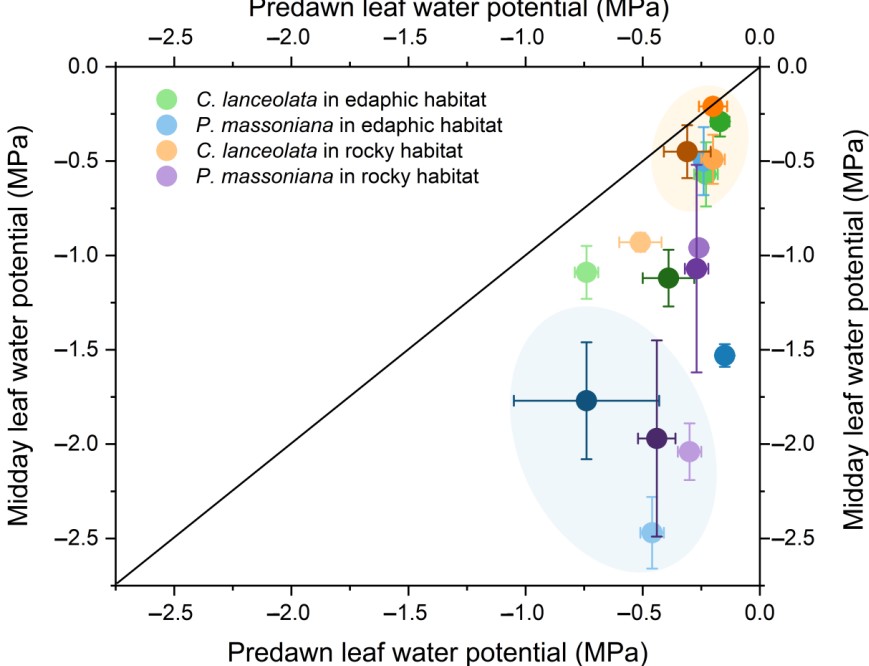

**Figure 8.** Plot of mean (±S.D.) predawn water potential and midday water potential of two tree species in the edaphic and rocky habitats. The black line is the 1:1 line of predawn water potential vs. midday water potential. The light to dark colors for each species represent the sampling dates as May-NP, June-HP, August-LP, and October-DP. The light orange shadow represents the cluster of trees close to the 1:1 line in HP and LP. The light blue shadow represents the cluster of trees away from the 1:1 line in NP and DP.

The diurnal ranges of water potential ($\Delta\psi$) exhibited significant variation in different species with seasonal changes ($p < 0.01$) (Figure 9). *C. lanceolata* showed significantly lower $\Delta\psi$ values than *P. massoniana* ($p < 0.001$). The $\Delta\psi_{max}$ value was the highest in NP for *P. massoniana* ($-1.84 \pm 0.19$ MPa) and in DP for *C. lanceolata* ($-0.45 \pm 0.34$ MPa). Both of the two tree species displayed the minimum $\Delta\psi$ ($-0.48 \pm 0.11$ MPa and $0.09 \pm 0.06$ MPa, respectively) in HP and LP. Both of the two species showed significantly higher diurnal ranges of water potential in the edaphic habitat than those in the rocky habitat ($p < 0.001$) during the sampling period, except for *P. massoniana* in LP and DP. Furthermore, there was no significant correlation between the $\Delta\psi$ values and water uptake depth for *C. lanceolata* or *P. massoniana* in the two habitats.

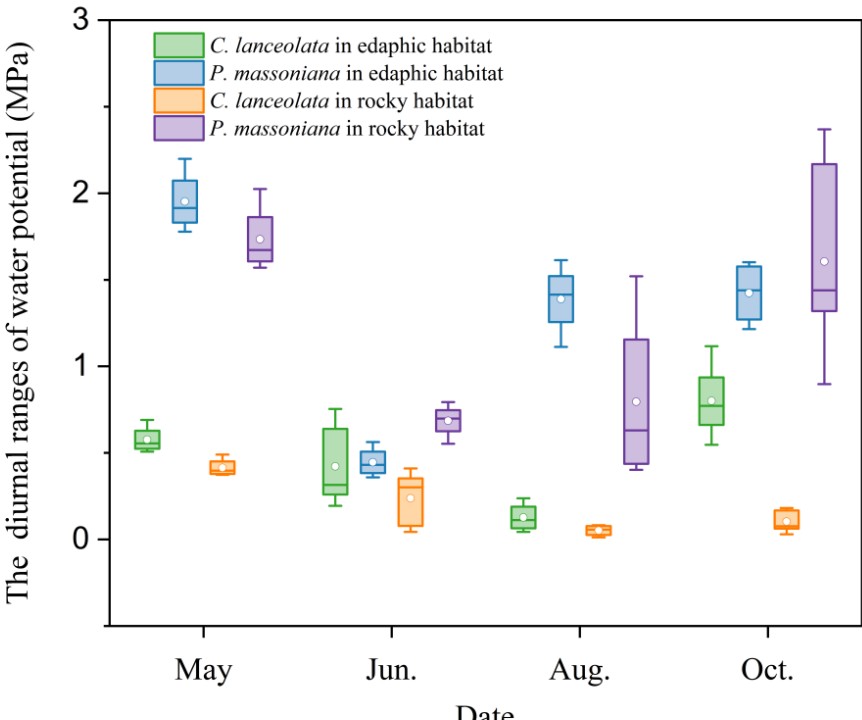

**Figure 9.** Variation in mean ($\pm$S.D.) diurnal ranges of water potential for *C. lanceolata* and *P. massoniana* during the wet season.

The high vapor pressure deficit and solar radiation showed a strong atmospheric evaporation force (AEF) (Table 2) that influenced transpiration and the diurnal changes in water potential. The meteorological factors during the sample period showed no significant variation ($p > 0.05$), except for DP with lower AEF. On the other hand the $\Delta\psi$ values exhibited seasonal changes and showed the highest values in NP and DP for the two species. These changes may be affected by the available of soil water. The responses of the $\Delta\psi$ to the precipitation amount ten days before sampling were different in the two tree species (Figure 10). The $\Delta\psi$ values of *C. lanceolata* did not increase with the change in conditions from no rain to high rainfall in the two habitats. However, the $\Delta\psi$ values of *P. massoniana* showed lower values with the precipitation increases in the edaphic and rocky habitats. Moreover, the plant water uptake depth was not correlated with the diurnal range of water potential (Table 3).

**Table 2.** The variations in meteorological factors during the sampling dates.

|  | T (°C) | RH (%) | VPD (KPa) | $PAR_{max}$ (umol·m$^{-2}$·s$^{-1}$) |
|---|---|---|---|---|
| May-NP | 26.6 | 63 | 1.29 | 1336.4 |
| June-HP | 30.1 | 74 | 1.11 | 1074.8 |
| August-LP | 31.1 | 66 | 1.54 | 1503.7 |
| October-DP | 20.9 | 68 | 0.79 | 1194.1 |

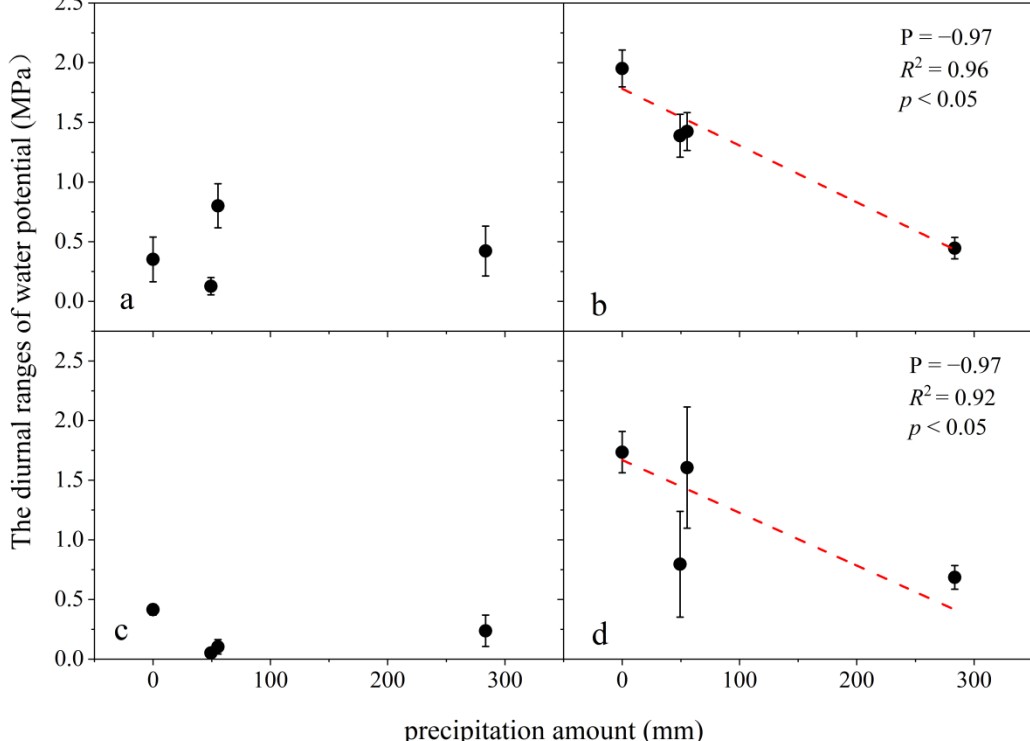

**Figure 10.** Relationships between the diurnal ranges of water potential (mean ± S.D.) and the precipitation amount ten days before sampling. P is Pearson correlation, $R^2$ represents the fitting degree of the relationship between the diurnal ranges of water potential and the precipitation amount; *p* is the *p*-value of the fitting ((**a**) *C. lanceolata* in the edaphic habitat; (**b**) *P. massoniana* in the edaphic habitat; (**c**) *C. lanceolata* in the rocky habitat; (**d**) *P. massoniana* in the rocky habitat).

**Table 3.** Relationship between water uptake depth and the diurnal ranges of water potential for *C. lanceolata* and *P. massoniana* in edaphic and rocky habitats.

| Pearson Correlation | Edaphic Habitat (−MPa) | | Rocky Habitat (−MPa) | |
|---|---|---|---|---|
| | *C. lanceolata* | *P. massoniana* | *C. lanceolata* | *P. massoniana* |
| Shallow layer | −0.41, $p > 0.05$ | 0.77, $p > 0.05$ | 0.328, $p > 0.05$ | −0.825, $p > 0.05$ |
| Middle layer | 0.152, $p > 0.05$ | −0.837, $p > 0.05$ | −0.411, $p > 0.05$ | −0.78, $p > 0.05$ |
| Deep layer | 0.441, $p > 0.05$ | −0.562, $p > 0.05$ | −0.064, $p > 0.05$ | −0.886, $p > 0.05$ |

### 3.4. Variation in Leaf Water Use Efficiency

The $\delta^{13}C$ values of the two species were significantly more negative in the middle wet season and early dry season than that in May-NP ($p < 0.05$) (Table 4). The leaf water use efficiency was higher in the drought stage in the wet season. There were no significant differences ($p > 0.05$) in $\delta^{13}C$ values between the two habitats. Furthermore, with the exception of in DP in the edaphic habitat and in LP in the rocky habitat, *C. lanceolata* and *P. massoniana* showed no significant differences in $\delta^{13}C$.

**Table 4.** Comparisons of $\delta^{13}C$ values of *C. lanceolata* and *P. massoniana* in edaphic and rocky habitats.

| | Edaphic Habitat (‰) | | Rocky Habitat (‰) | |
|---|---|---|---|---|
| | *C. lanceolata* | *P. massoniana* | *C. lanceolata* | *P. massoniana* |
| May-NP | −25.78 ± 0.85Aa | −26.67 ± 0.73Aa | −25.66 ± 0.57Aa | −26.93 ± 0.05Aa |
| June-HP | −27.07 ± 0.41Ba | −27.75 ± 0.85Ba | −27.95 ± 1.26Ba | −28.31 ± 0.18Aa |
| August-LP | −27.65 ± 0.69Ba | −28.8 ± 0.42Ba | −27.43 ± 0.18Ba | −29.51 ± 0.27Bb |
| October-DP | −27.22 ± 0.47Ba | −28.23 ± 0.18Bb | −27.86 ± 0.64Ba | −28.86 ± 0.25Ba |

Note: Capital letters represent significant differences of the same tree species among different sampling dates at the 0.05 level; lowercase letters represent significant differences between *C. lanceolata* and *P. massoniana* in the same habitat at the 0.05 level.

## 4. Discussion

### 4.1. Water Uptake of Tree Species in Two Habitats

The variation of plant water uptake depth in the two habitats was consistent, except in June-HP. These two species, growing at the foot of the slope, mainly absorbed soil water from shallow layers in the early and middle wet season, and switched to deeper layers in the late wet season. This water uptake pattern has also been observed in other natural species and plantations in the similar study areas [16,28]. However, it was noted that the plants utilized shallow soil water rather than deep water (no springs flowing) in the early wet season with a 20 day drought, which was inconsistent with other studies in this climate region [16,22]. Although the mean soil moisture was lower compared to other samplings, the VWC was still higher than that observed in semiarid climate regions in the wet season [15,43]. Meanwhile, with a relatively lower wilting coefficient and high spatial heterogeneity [44], the shallow layers could also provide enough available water for plants. Previous studies showed that the plant species adjusted their physiological factors, such as water potential behavior, water use efficiency, in response to the environment changes [25,45,46]. In our study, *C. lanceolata* and *P. massoniana* exhibited the highest leaf $\Delta\psi$ and $\delta^{13}C$ values in NP, indicating that they tried their best to absorbed enough shallow soil water with lower midday leaf water potential to balance carbon-water relations in tandem with high leaf-level intrinsic water use efficiency (iWUE). Moreover, this water uptake pattern is an adaptation, enabling plants to save more energy for growth in the early wet season. Both *C. lanceolata* and *P. massoniana* grow quickly, showing high energy consumption in May, as well as in the early growing season. Although the deep soil layer has a higher VWC, the energy required to take up water from the deep layer is greater than that of the upper layers [47,48]. Thus, the trees extracted shallow soil water to avoid excessive energy consumption through physiological adjustments [49–51]. In the middle and late wet season, plants water uptake depth shifted from shallow to deeper layers. Soil water availability may be the main reason for this water uptake pattern [52,53].

When the antecedent precipitation was much higher in the middle wet season, the plants still absorbed water from shallow layers in the rocky habitat, but in the edaphic habitat they switched to deep layers of soil water. Water availability is the most important factor influencing the plants water uptake depth [26,54]. Soil structure, such as soil texture, bulk density, affected water holding capacity, and migration, along with soil profiles, thus regulated plant water use [28,55]. The bulk density in the rocky habitat was lower than that in the edaphic habitat, promoting the high water holding capacity, whereas in the thin deep layers with large cracks and crevices in the rocky habitat, moisture leaked into the layer, flowing through the springs. In the thick deep layer with fine cracks in the edaphic habitat, the amount of stored water was higher than that in the shallow layer after large and continuous precipitation. Therefore, discrepancies in soil properties are the main reasons underlying the different soil water availability along the profiles in the two habitats. Furthermore, the low diurnal ranges of water potential of *C. lanceolata* and *P. massoniana* also demonstrated that they were both had sufficient water supplies in the two habitats.

*4.2. Water Uptake Pattern and Physiological Factors Change in the Different Tree Species*

The two coexisting plants—both in the edaphic and the rocky habitat—exhibited no significant differences in water uptake patterns, indicating that they had the same eco-hydrological niche and no water source segregation. This result was consistent with a previous study in a similar climate region, which showed that the six mixed plantations had similar water sources, using the 0–30 cm soil layers in the wet season [28]. Studies in other regions also showed that coexisting species usually exhibited water competition in mixed stand [56,57]. Nie [58] investigated three communities on adjacent rocky hill slopes, and found that different species within each community all exhibited the use of a similar water source. Du [59] studied three karst climate forest communities of a typical hill, and obtained the same results. The similar root distribution of *C. lanceolata* and *P. massoniana* may be the main reason that they exhibited the same water uptake pattern [60,61]. Hence, the interspecific difference in community was relatively low in the subtropical monsoon climate region. However, as per the above analysis, the water uptake pattern was different between the edaphic and rocky habitats for the same species. This suggested that the habitats may have more of an influence on plant water use than the interspecific differences in the community, especially when the antecedent precipitation is high.

Although the water uptake depth was similar for the two species, the two species had different physiological responses to the water uptake. In our study, *C. lanceolata* maintained small diurnal ranges of water potential, high leaf $\delta^{13}$C values, and a large amount of branching from the base of the trunk, whereas *P. massoniana* showed the inverse characteristics. Meanwhile, the $\Delta\psi$ values of *P. massoniana* in the two habitats were negatively associated with antecedent precipitation, but a significant relationship was not observed in *C. lanceolata*. Wang [15] found similar results for two species in a mixed plantation in the Loess Plateau. Moreno-Gutiérrez [50] observed the existence of species-specific eco-physiological niche segregation in dryland plant communities. A possible explanation was that the inter-specific competition in the same habitat caused each tree species to establish different hydrological niches for water uptake [48,62]. Unlike the previous findings, in our study, there was significant eco-physiological niche segregation but no ecohydrological segregation for the two species in the same habitat. The plant water uptake depth was not correlated with the diurnal range of water potential. In other words, the aboveground physiological parameters showed significantly differences between two species, whereas the belowground water uptake was consistent among the two species. This discrepancy may be attributed to sufficient precipitation and soil water availability for ecohydrological non-segregation [63] and interspecific differences in terms of eco-physiological segregation [54].

*4.3. Implications for Plant Water Adaptation under Precipitation Changes*

With the increasing temperatures, precipitation patterns change seasonally and become more variable [8], which could lead to an increase in either the severity of drought or extreme precipitation, especially in the growing season [64–66]. When drought or extreme precipitation occurs, soil water availability may influence the plants' water use strategies.

In our study, plants absorbed soil water from shallow layers by increasing the diurnal ranges of water potential and water use efficiency in the early wet season with a 20 day drought. The tree species sought a balance between water uptake and growth through their relatively high water use efficiency [67]. However, if the drought was prolonged, soil moisture would decline and fail to supply water for plants. Ding [26] conducted a 135 day rainfall exclusion experiment in a catchment, and found two adverse responses, according to different physiological characteristics, to the severe water limitation: canopy defoliation and/or mortality and survival. In our study, *P. massoniana*, as the species exhibiting profligate water use exhibited larger $\Delta\psi$ and lower $\psi_{\mathrm{md}}$ values for absorbing water sources [26]. Once the $\psi_{\mathrm{md}}$ values beconmes lower than the hydraulic trait values, the species may suffer from the risk of hydraulic failure, such as xylem cavitation and leaf turgor loss [56,68]. On the contrary, *C. lanceolata* displayed stable $\Delta\psi$ values in the

sampling period, indicating the rigorous stomatal control [69]. The tree growth rate of *C. lanceolata* may slow due to the reduction in shallow soil water sources and the advanced stomatal closure.

Except for the drought in the growing season, the frequency of rainstorms and extreme precipitation also has also been increasing recently [70,71]. Plants are the main conduit for returning terrestrial water to the atmosphere, thereby exerting a strong effect on hydrologic fluxes of the terrestrial-atmospheric system [63]. In our study, the plants that mainly utilized for deep layer soil water in the edaphic habitat and the $\Delta\psi$ values of *P. massoniana* were lower when the precipitation was extremely high. These results illustrated that the tree species could adjust their water use strategies and increase the eco-hydrological linkages between the ecosystem and the deep-layer system [59].

## 5. Conclusions

In this study, the stable isotope technique and a pressure chamber were applied to detect the seasonal water uptake patterns of two coniferous species in edaphic and rocky habitats in a subtropical monsoon climate region. The results showed that the two species mainly absorbed soil water from shallow layers, even in the short drought period in the wet season and switched to deeper layers in the early dry season. It was noted that the trees utilized deep-layers water in edaphic habitats when the antecedent rainfall was high. The two species showed no significant differences in water uptake depth, but notable differences in their leaf water potential behaviors. *C. lanceolata* displayed narrow and stable $\Delta\psi$ values whereas the $\Delta\psi$ values of *P. massoniana* were negatively associated with antecedent precipitation. Thus, for co-existing species in communities, there was significant eco-physiological niche segregation but no eco-hydrological segregation. For the tree species in the two habitats, the water uptake depth was influenced by the soil water availability, but the physiological factors were unchanged, determined by the species genetics. Furthermore, during a long drought in the growing season, *C. lanceolata* and *P. massoniana* probably show divergent responses, such as growth restriction and hydraulic failure. However, when the precipitation is heavy and long, these species could increase the ecohydrological linkages between the ecosystem and the deep-layer system in the edaphic habitat.

**Author Contributions:** J.J. and Y.C. conducted field experiment, performed data analysis, and wrote the draft manuscript. J.L. and W.Y. conceived the study, designed the experiment. All authors contributed to discussion and interpretation of resulting data. All authors have read and agreed to the published version of the manuscript.

**Funding:** This research was funded by the key research and development program in Hunan province (2020NK2022), the National Natural Science Foundation of China (41807162) and the Hunan Province Natural Science Foundation (2019JJ50994).

**Data Availability Statement:** All relevant data are within the manuscript.

**Conflicts of Interest:** We declare that we have no financial and personal relationships with other people or organizations that can inappropriately influence our work, there is no professional or other personal interest of any nature or kind in any product, service and/or company that could be construed as influencing the position presented in, or the review of, the manuscript entitled "Water uptake pattern by coniferous forests in two habitats linked to precipitation changes in subtropical monsoon climate region, China".

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
