# Peer review of "Water Uptake Pattern by Coniferous Forests in Two Habitats Linked to Precipitation Changes in Subtropical Monsoon Climate Region, China"

_forests, doi:10.3390/f13050708_

Round 1

Reviewer 1 Report

The paper presents how the relative uptake of water from the different soil depths changes during the wet season in two species. The analysis of stable isotopes is complemented with measurements of water potential. The paper generally reads well. 

My main objection is that the study was conducted only in the wet season. Therefore, the water content does not much change between the soil depths, and there is almost always a lot of water (i.e. no water stress). It is not entirely clear, how from this data authors managed to estimate the water uptake by depths. No data on stable isotopes of hydrogen and oxygen in the xylem are presented. These two points shall be significantly improved in the revised version of the manuscript.

The part on water potential is written with less rigor compared to the part on the isotopes. The presentation in graphs and the structure of this part may be improved. There are not many links in the text between isotopes and the water potential. 

The discussion is sometimes quite speculative and may be shortened. Table 3 shall be removed from the discussion to the results. 

The specific comments are in the attached PDF.

Author Response

Thank you for these appreciative and encouraging comments, which  were all valuable and very helpful for revising and improving our paper. Please see the attachment of the point-by-point response to the reviewer's comments.

Reviewer 2 Report

This study combined water isotope and physiological measurements to quantify ecohydrology processes of two main conifers in subtropical China. The authors found that plant water uptake is homogeneous between habitats, and water uptake depths switched from shallow soil layers in early-season dry period to deep ones during the late wet season with amply precipitation. The findings are timely and will provide insights into forest management in subtropical China, where climate extreme is increasing in recent decades. However, the whole text needs to be thoroughly edited and checked. In addition, the language editing by a native speaker is required.

L31-32, which species respectively show growth restriction and hydraulic failure?

L54-55, Tree species may switch their water sources from shallow layers in wet season with sufficient precipitation to stable layers in dry season.

L59, Other plants, however, mostly…

L62, characteristics

L79, Subtropical China characterized by a monsoon climate is an ecologically sensitive area under global change.

L87-90, While the authors stated the importance of investigating plant water uptake in different habitats, what are the specific differences (elevation, soil type or tree species etc.) in Habitat was not introduced and why they are important for vegetation restoration.

L92, I would suggest summarizing the related researches using isotopic methods to quantified plant water uptake (at least in the study areas).

L95-102, The Objectives and Hypotheses are more general, which need to be improved.

L108-110, The wet season, which receives more than 60% of annual rainfall, lasts from late April to late September, and the dry season extends from December to February.

L118-130, As authors stated in the method, the habitats are mostly related to soil type. However, to make readers more easily understand the contrasting habitats, habitat photographs and its spatial distributions in the studied area need to be added in Figure 1.

L157-158, Did the DBH of sampled trees represent the average DBH in the stands?

L162-166, Citations are required.

L175, Information soil bulk density are missed here.

L209, what is the meaning of ‘2/3 of the way up of the canopy’?

L217-218, I doubt the statements of no isotopic discrimination of water between plant and soil water, especially for the current studied species.

L230, what is ‘no effective rainfall’?

L263, I would suggest insetting date in each panel rather than in the caption.

L270, Did the average VWC differ significantly between habitats.

L283-284, The meaning of this sentences is unclear.

L282-293, This paragraph and elsewhere in RESULTs needs to be rewritten to convey the findings in logical ways. According to Figure 6, the water uptake depth did not differ between species irrespective of the habitats, whereas they did differ between habitats and across seasons.

L330-339, What is ‘diurnal range of water potential’? Diurnal variations in water potential may also affects by atmospheric drought (i.e., VPD). In fact, according to Table 2, plant water uptake depth is not correlated with diurnal range of water potential.

The relatively roles of VPD and soil drought on plant water status should be quantified.

L341, please explain Rda. and Roc in Figure 7.

L353, define shallow, middle and deep layers.

L375 &L379, delete ‘For one thing’ and ‘For another’.

L394, wrong unit for carbon isotopic values in Table 3.

L383-390, The authors attributed plant water uptake from shallow soil depth to energy saving strategies. However, direct links between measured iWUE and plant water uptake depth are missing according to the current measurements and explanations.

L435, change ‘different’ to ‘difference’

L442-445, Again, the contrasting eco-physiological between species may cause by atmospheric drought.

L463, this is a speculation, not the finding of this study.

Author Response

Thanks for the valuable and helpful comments. We have revised carefully and please see the attachment in details.

Round 2

Reviewer 1 Report

Thank you for the revision.

Reviewer 2 Report

This revision have been greatly improved. I do not have any further comments or suggestions.